# SETTING THE DC: TOOL-GROUNDED D&D SIMULATIONS TO TEST LLM AGENTS

## ABSTRACT

Dungeons and Dragons (D&D) has been considered to be an intellectually challenging game for strategy planning and role-playing. Large language models (LLMs) are increasingly deployed as autonomous or semi-autonomous agents, yet most evaluations still target single-turn QA or short-horizon tasks. Assessing agentic performance in rules-constrained, multi-step settings is challenging because style-conforming narration can diverge from task optimality. In this work, we present D&D Agents, a benchmark built on a multi-agent Dungeons & Dragons simulator. In our benchmark, LLMs use tools to query and update the game state, assuming the roles of referee ('Dungeon Master', DM), players, and adversarial monsters in tactically rich combat. This benchmark setting requires long-horizon planning, compliance with game rules, varied agent personas, and grounded interaction with the game state. We evaluate transcripts and tool traces along six axes—Function Usage, Parameter Fidelity, Acting Quality, Tactical Optimality, State Tracking, and Function Efficiency—capturing both capability and reliability in closed-loop play. Our benchmark allows researchers to run identical seeded scenarios with auditable traces, making error analysis and algorithmic improvements (prompting, tool-use policies, memory) straightforward and comparable.

## 1 INTRODUCTION

Large language models (LLMs) are increasingly deployed as tool-using agents that must plan over long horizons, remember salient context, and coordinate with other actors. Early benchmarks emphasize single-agent or short-horizon QA, leaving open how to *evaluate* memory, planning, and coordination in settings where natural language drives perception and intent but *rules* govern what actions are legal (Li et al., 2023; Wu et al., 2023; Du et al., 2023). Work on self-reflection and persistent memory suggests paths to stabilize behavior over many turns (Shinn et al., 2023; Park et al., 2023; Li & Gupta, 2025), but we still lack testbeds that expose the full tangle of multi-step planning, strict rule adherence, and team strategy.

We argue that **Dungeons & Dragons (D&D)** is a natural evaluation ground for these skills: an initiative-driven, mixed cooperative–adversarial game where agents must remember evolving state, communicate succinct plans, and translate intentions into rule-compliant actions. Crucially, D&D couples *team coordination* with *opponent-aware tactics* under partial observability, a bounded action economy, and spatial constraints with stochastic resolution—collectively yielding a non-stationary multi-agent setting that stresses planning, memory, and communication. Because play unfolds through dialogue, D&D also opens a direct avenue for *human–AI interaction*: agents can assist or co-play with people, and the same mechanics support scalable evaluation of agent decisions.

In this work, we present D&D Agents, a novel multi-agent simulation framework in which LLM-driven agents assume the roles of DM, players, and monsters to autonomously play out tactically rich D&D combat encounters. This framework serves as both a research environment – capturing the complexities of autonomous agent evaluation, long-horizon rule-following behavior, and multi-agent coordination – and as a testbed for new methods to ground LLM decisions in a formal game system. D&D Agents comprises a high-fidelity simulator and a suite of tools that bridge natural language and game mechanics. Through careful prompt design, we imbue each agent with a distinct role and objectives. We pair our environment with a six-axis metric suite and validate our automatic

judges against human ratings, finding strong alignment (Pearson $r \approx 0.96$–$0.98$); for example, the judge's means closely track human means–Acting 0.572 vs. 0.601 and Tactical 0.551 vs. 0.568–supporting credible large-scale assessment.

Our main contributions are summarized as follows:

1. We develop a fully automated D&D combat simulator where multiple LLM agents engage in battle under authentic game rules. This is the first framework to pit LLM "players" against an LLM "Dungeon Master" in a closed-loop environment that rigorously enforces turn-based game mechanics and stochastic outcomes (dice rolls). It also supports human-AI co-play–People can assume any subset of player roles (from zero to all) while the remaining roles are controlled by LLMs.

2. We design a structured API of game actions, each with predefined parameters and precondition checks, to ground the agents' decisions. This approach cleanly separates narration from mechanics: the DM agent may describe events in natural language, but the truth of those events is guaranteed by the underlying tool calls.

3. We introduce a prompting scheme that guides the DM and player agents to fulfill their in-game roles. This scheme enables multi-agent coordination and opposition purely through learned communication and tool use, without any hard-coded game logic.

4. To rigorously evaluate the performance of our D&D Agents, we define six evaluation axes that capture both the capabilities and reliability of the agents in long-horizon gameplay. We evaluate transcripts and tool-call traces along these dimensions to quantify progress and identify failure modes in an objective, reproducible manner.

## 2  RELATED WORK

A growing line of work grounds language agents in *executable* interfaces so long-horizon behavior is less ambiguous and more auditable. Programmatic tool use—via function calling or API invocation—improves reliability in interactive environments (ReAct; Toolformer; MRKL; Gorilla) (Yao et al., 2022b; Schick et al., 2023; Karpas et al., 2022; Patil et al., 2023), and similar constraints help in text games and web tasks (Jericho/interactive fiction, ALFWorld, ScienceWorld, WebShop, WebArena) (Hausknecht et al., 2020; Shridhar et al., 2020; Wang et al., 2022; Yao et al., 2022a; Zhou et al., 2023) as well as open-ended game worlds like Minecraft (Voyager; MineDojo) (Wang et al., 2023; Fan et al., 2022). These results suggest that defining a compact, typed action space is a practical route to robust multi-step agents.

Within D&D, prior work treats gameplay primarily as *dialogue and state tracking*. Callison-Burch et al. (2022) frame D&D as a dialogue/state challenge; FIREBALL provides actual-play transcripts with structured state and executable Avrae commands (Zhu et al., 2023b); CALYPSO and Overhearing explore DM assistance tools (Zhu et al., 2023a; 2025). However, these systems typically operate on a *single player at a time* and are not closed-loop multi-agent simulations across many turns; moreover, the *game mechanics are fully simulated in handwritten code* (e.g., Avrae), with the LLM advising rather than executing mechanics. Complementary efforts outside D&D explore multi-agent interaction in rule-based environments (Thudium et al., 2025) and LLM-driven *game simulation* more broadly (Song et al., 2024), reinforcing the value of structured interfaces for coordination and competition.

Our work differs in placing LLMs *directly* in the loop as DM and multiple players within a rules-enforcing simulator: every effect-producing action is executed via a typed API, producing deterministic, auditable traces. This enables closed-loop, turn-by-turn evaluation of cooperation and opposition among multiple agents, supports human co-play, and yields standardized, seedable scenarios for fair comparison.

## 3  SIMULATION FRAMEWORK

**State.** The state consists of two main components: (i) *Character creation* and (ii) *map generation*. We implemented a structured character creation system that uses LLM agents to generate D&D 5e player and monster characters via AI function calls. The `CreatePlayerKani` agent

Table 1: LLM failures in simulation. **Top:** The model checked that there is no sight between the player and the monster, but still attempted to attack. **Bottom:** The model checked the enemy's HP to be 0 (dead), but did not attack other enemies.

| | |
|---|---|
| *Player*: **Elaria**    *Enemy*: Goblin 1 | |
| **Elaria**:    Thinking...    `[check_valid_attack_line(attacker_name='Elaria', defender_name='Goblin 1')]` | |
| **System**: FUNC: `False` | |
| *Model Response* | |
| **Elaria**: I will attack Goblin 1 with my short bow. `[roll_attack(attacker_name='Elaria', defender_name='Goblin 1', roll_type='normal', modifier=4, weapon_name='Short Bow', action_cost=1)]` | |
| *Player*: **Briana**    *Enemies*: Goblin 1, Goblin 2 | |
| **Briana**: Thinking... `[check_hp(name='Goblin 2')]` | |
| **System**: FUNC: `0` | |
| *Model Response* | |
| **Briana**: Excellent! I have a clear shot. I'll take aim and burst my firebolt at Goblin 2! | |

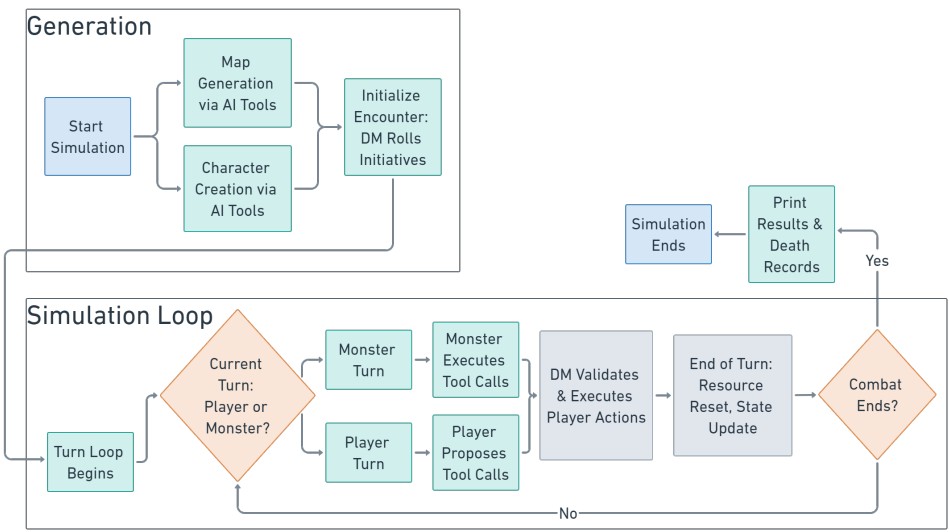

Figure 1: The simulation framework contains two major components: The generation step (Top) and the simulation step (Bottom). Background settings are generated in the generation step, while LLM/human players can take turns in the simulation loop to execute actions.

prompts the model with official creation rules and user input to generate legal characters, while `CreateMonsterKani` uses official monster data to instantiate enemies. External D&D APIs provide canonical resources, and derived properties are automatically computed according to rules. For spatial context, we provide two seedable map modes that yield traversable, height-aware grids. Indoor maps are rasterized from compact JSON layouts (rooms, walls, doors), while outdoor maps are procedurally generated to ensure connectivity with distant start/end anchors. Both encode discrete height values for slope-aware movement and use line-of-sight checks to gate ranged actions. A fixed seed ensures reproducibility.

**Actions.** The simulator exposes a typed API of deterministic function calls that define the action space. Calls are validated against preconditions (initiative ownership, budgets for action/bonus/reaction/movement, spell slots, range, line of sight, target existence, status effects). We group functions into six categories: 1) *Query/validation* (state checks, LoS tests); 2) *Movement/positioning* (move, dash, disengage); 3) *Dice primitives* (roll_dice); 4) *Attack/spell resolution* (roll_attack, roll_save,

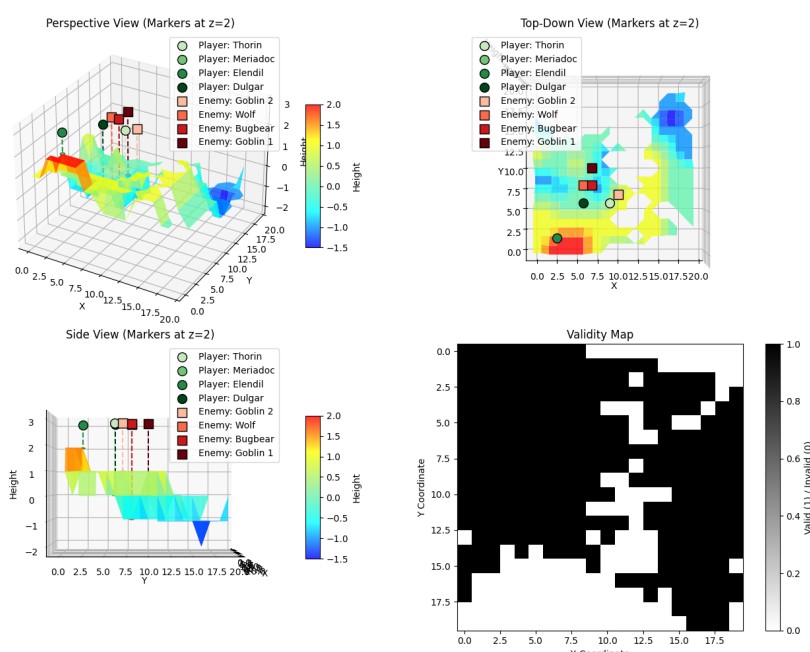

Figure 2: An outdoor map showing current alive character positions.

roll_dmg); 5) *Turn economy/bookkeeping* (roll_initiative, reset_resources, check_concentration); 6) *Rendering* (visualize_map).

**Transition Dynamics.** Function calls are atomic and deterministic given sampled dice rolls. The simulator enforces legality and automatically updates resources, HP, position, or conditions. A turn consists of querying state, executing movement or attacks, and concluding with resource resets and audits. Figure 1 provides an overview of the simulation.

**Observations.** Agents observe a combination of natural language narration and structured returns from simulator functions (e.g., query results, dice outcomes). Maps can be visualized after each move, and observations are local to the calling agent, yielding partial observability. Figure 2 is an example map generated by the map generator about a combat scene between four players and four enemies.

**Reward.** For evaluation, we measure downstream combat outcomes and auxiliary metrics such as efficiency of function usage and error rates. When used for MARL, task-specific rewards can be shaped around these signals.

**DM Agent.** The DM is an LLM steered by GM_PROMPT that behaves like a transactional controller: it plans in natural language but *executes* through a small, typed set of AI functions with validation, atomicity, and explicit bookkeeping. In play, it follows a fixed recipe: *query - (optional) move - validate - resolve - bookkeep*, rolling and announcing initiative with roll_initiative; on each turn it queries state, moves with move when needed, gates ranged options via check_valid_attack_line, resolves attacks/spells (roll_attack, roll_spell_attack, roll_save, roll_dmg), applies HP/resource updates, audits temporary conditions/resistances/concentration, and finishes with reset_resources and reset_speed, emitting <End Turn/>.

The prompt functions as a declarative control policy: narration is descriptive while functions are authoritative; explicit if–then gates (range/LoS/reach/resources/economy) prevent illegal actions and route failures to repairs (reposition, alternate action, end turn); parameters must come from canonical sources; economy semantics for *Dash/Disengage* are tied to budgets; and within-turn caching improves efficiency. It also installs stable event handlers (e.g., opportunity attacks on leaving reach), compact zero-shot tactical heuristics, and archetypal exemplars (single-target attack-roll, save-based

AoE) that generalize to unseen abilities; a small condition glossary enables status handling without bespoke code. Optimized for adherence with a concise "contract," exact verb–function alignment, and a numbered end-of-turn checklist capped by a sentinel token, this design yields consistent, rules-compliant, and *auditable* tool-call traces across models while remaining portable and easy to extend.

**Player Agent.** The player agent is an LLM guided by `PLAYER_PROMPT` that converts tactical intent into concrete, legal actions for its character while coordinating with allies. In the playthrough it follows a *sense - plan - validate - act - communicate* routine: (i) at turn start, it queries state and resources; (ii) selects movement and economy modifiers consistent with budgets; (iii) for ranged options, first gates with `check_valid_attack_line` and computes distance/reach from the queried positions; (iv) specifies its chosen action (attacks/spells), invoking simple query functions directly but proposing functions which change the game state for the DM to execute to avoid hallucination and parameter fidelity; and (v) emits concise narration and optional team messages to coordinate surround an enemy (flank), focus multiple allies on one target (focus fire), or pull pressure off an ally (peel). The DM remains the authoritative executor—committing any state-changing calls and running the end-of-turn checklist—which grounds player intent and yields an auditable tool-call trace aligned with the transcript.

The `PLAYER_PROMPT` emphasizes intent expression and cooperation under uncertainty rather than adjudication. It instructs the agent to *ask or check* when unsure about geometry, reach, or spell parameters, preventing silent errors while keeping turns efficient. Narration is kept concise and role-separated: one–two sentences to summarize intent/outcomes, with coordination messages isolated from flavor so allies (and the DM) can parse plans quickly. A lightweight direct-message protocol—`<Call/>`Name, Message`<Call/>` with strict formatting—provides a reliable, code-free communication channel; concrete templates (e.g., chaining actions, requesting healing) enable accurate addressing and improve teamwork (timed flanks, handoffs, prioritized healing), while all mechanical effects remain confined to AI functions executed by the DM.

## 4 EXPERIMENTS

**Evaluation settings.** We use 27 seedable scenarios packaged as save JSONs, constructed by a $3 \times 3 \times 3$ design: three four-class character groups $\times$ three stat tiers (low/medium/high) $\times$ three monster-map sets. Across the three groups, all 12 core D&D classes are represented. Each monster–map set has a custom enemy roster from three well-known fantasy skirmish set-ups (from 'Lost Mine of Phandelver'): Goblin Ambush, Kennel in Cragmaw Hideout, and Klarg's Cave (Wizards RPG Team, 2014). All models run on the identical 27 files; no per-model tuning of maps, parties, or monsters is permitted. Each episode lasts ten turns, after which we export the dialogue transcript and the ordered tool-call trace; these artifacts feed our six metrics–Function Usage, Parameter Fidelity, Acting Quality, Tactical Optimality, State Tracking, and Function Efficiency. We test *Claude Haiku 3.5*, *GPT-5* (OpenAI, 2025), *DeepSeek V3.1* (Liu et al., 2024), *Qwen3-32B (base)* (Team, 2025), *Qwen3-235B-A22B-Thinking-2507 (thinking)*, *Qwen3-Next*, and *GLM-4.5-Air* (Zeng & Team, 2025). We adapt a role-swapping copilot protocol in which DeepSeek V3.1 fills the complementary side: when a target model is evaluated as DM and monsters, DeepSeek V3.1 plays the players; when the target model is evaluated as players, DeepSeek V3.1 plays the DM and monsters.

**Function and function parameter efficiency.** We evaluate function calling performance across 27 combat scenarios using both automated log-derived metrics and human evaluation. The automated evaluation identifies incorrect function calls (improper function selection resulting in execution errors) and incorrect parameter usage. Human evaluation additionally assesses incorrect function selection that does not trigger execution errors, missing function calls (false negatives), and extraneous function calls (false positives), as summarized in Table 2, 3.

We found that models with smaller parameters have significantly lower DM performance than the player. *Qwen3-32B (base)* has 20% incorrect function calls when acting as a DM, but only 4.58% incorrect function calls when acting as a player. This shows that models have a role-specific deficit because of context load and tool routing. Acting as DM requires carrying the longest working set. In our framework, the DM is the executor that plans in language but must commit mechanics through typed API calls, with strict preconditions. This expands the token/context that the model must handle every turn and increases the chance of routing or parameter errors.

Table 2: Automated function-use correctness and efficiency. We use a log-based checker to automatically find incorrect function usage and incorrect parameter usage (lower is better). We then use the log to identify incorrect function selection, missing and unnecessary calls, and F1 against gold plans.

| Model | Incorrect func (%) | Incorrect params (%) | Incorrect Selection (%) | Missing (%) | Unnecessary (%) | F1 (%) |
|---|---|---|---|---|---|---|
| DeepSeek V3.1 | 3.15 | 2.47 | 1.79 | 28.99 | 1.86 | 80.61 |
| GPT-5 | 2.84 | 2.38 | 1.46 | 11.27 | 1.24 | 91.51 |
| Claude Haiku 3.5 | 1.17 | 1.14 | 0.55 | 6.83 | 1.01 | 95.18 |
| GLM-4.5-Air | 3.44 | 1.62 | 1.90 | 20.40 | 1.50 | 86.40 |
| Qwen3-235B-A22B-Thinking-2507 | 3.57 | 1.44 | 2.00 | 24.75 | 1.42 | 84.02 |
| Qwen3-32B (base) | 12.40 | 4.85 | 5.20 | 58.60 | 3.90 | 54.11 |
| Qwen3-Next | 9.96 | 3.70 | 4.10 | 49.30 | 3.35 | 61.72 |

Table 3: Human-evaluated function-use correctness and efficiency. Human annotators find the incorrect function usage, incorrect parameter usage, incorrect function selection, missing, and unnecessary calls based on the pipeline given in the prompt. Finally, an F1 is calculated against gold plans.

| Model | Incorrect func (%) | Incorrect params (%) | Incorrect Selection (%) | Missing (%) | Unnecessary (%) | F1 (%) |
|---|---|---|---|---|---|---|
| DeepSeek V3.1 | 3.09 | 2.53 | 1.74 | 28.24 | 1.82 | 78.81 |
| GPT-5 | 2.77 | 2.43 | 1.50 | 10.96 | 1.21 | 93.59 |
| Claude Haiku 3.5 | 1.14 | 1.16 | 0.56 | 7.01 | 0.98 | 92.35 |
| GLM-4.5-Air | 3.51 | 1.67 | 1.95 | 19.98 | 1.46 | 83.93 |
| Qwen3-235B-A22B-Thinking-2507 | 3.49 | 1.48 | 2.04 | 25.30 | 1.46 | 81.83 |
| Qwen3-32B (base) | 12.09 | 4.75 | 5.32 | 57.03 | 3.81 | 52.59 |
| Qwen3-Next | 10.19 | 3.59 | 4.19 | 50.48 | 3.42 | 60.33 |

**State-Tracking Accuracy.** We assess state-tracking accuracy to measure whether agents maintain coherent internal representations of game state throughout scenario execution. Here, we specifically target hallucination errors where models generate actions inconsistent with established game state, such as attacking with weapons not present in inventory or referencing non-existent status effects. We break the error type to four different error types:

- Status Effect Errors: Claiming buffs/debuffs that weren't applied or ignoring active conditions

- Positional Inconsistencies: Misremembering character locations, movement capabilities, or terrain features

- Resource Tracking: Incorrect HP, using non-existent items, or action point calculations

- Entity State Confusion: Mixing up which characters are alive/dead, conscious/unconscious

The error rate is shown in Table 4. We also created a turn-based error rate analysis in Figure 3. Although there are only very few entity state actions, they represent a considerable source of hallucination errors across all models. Since entity state error only happens in the late state of the game log, after removing it, the temporal analysis still indicates that hallucination rates increase sharply at turn 2 and fall continuously with scenario length in all models. This might be because turn 2 is the actual turn that initiates the simulation, and all models have limited knowledge of the environment, which can lead to an increasing hallucination rate. The decreasing hallucination rate is a sign that all models are adapting to the task in a longer context.

**Acting Quality.** We assess how well models stay "in character" and write natural action beats across 27 combat scenarios. For each scenario we first keep only narrative sentences (speaker text, not DM/tool output), filtering out digits and dice notation. Each remaining sentence is labeled persona if it shows a recognizable voice or in-world action beat-via speaker-specific cues (e.g.,

Table 4: State-tracking error rates by category across models. The error rate is calculated by the total error in this category divided by the total number of actions.

| Model | Status Effect | Positional | Resource | Entity State | Total |
|---|---|---|---|---|---|
| DeepSeek V3.1 | 0.173 | 0.006 | 0.064 | 0.384 | 0.043 |
| Claude Haiku 3.5 | 0.098 | 0.000 | 0.034 | 0.107 | 0.010 |
| GPT-5 | 0.111 | 0.001 | 0.041 | 0.184 | 0.020 |
| GLM-4.5-Air | 0.156 | 0.002 | 0.058 | 0.258 | 0.031 |
| Qwen3-235B-A22B-Thinking-2507 | 0.177 | 0.002 | 0.065 | 0.292 | 0.037 |
| Qwen3-Next | 0.273 | 0.004 | 0.138 | 0.616 | 0.086 |
| Qwen3-32B (base) | 0.340 | 0.004 | 0.162 | 0.726 | 0.103 |

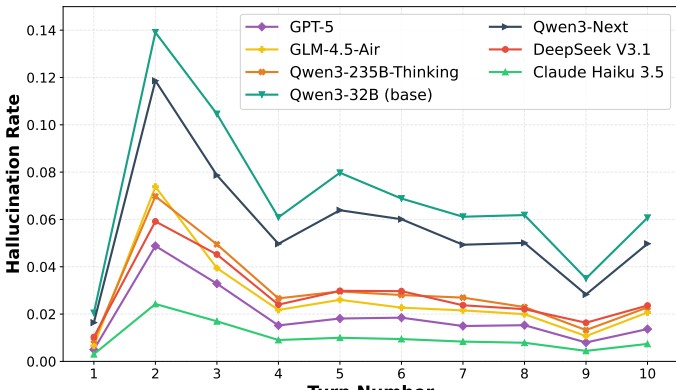

Figure 3: The hallucination rate of the model calculated by total hallucinated actions / all actions. We removed entity state errors here, as most entity state checking occurs only in the late game.

paladin (armored melee) valor, ranger (archer-scout) poise, warlock (occult caster) edge, druidic (nature caster) calm, monster taunts/imperatives); first-person physical action beats also count. The scenario score is

$$A = \frac{1}{2} \frac{S_{\text{persona}}}{S_{\text{narr}}} + \frac{1}{2} \min\left( \frac{T_{\text{distinct}}}{T_{\text{max}}}, 1 \right)$$

where

$$T_{\text{max}} = \left( N_{\text{player characters}} + N_{\text{monster types}} \right) + 1$$

Thus, $A$ balances how often the writing feels in-character (persona density) with how many different voices the model sustains (trait coverage). To validate the automatic Acting Quality metric, we ran a human evaluation on 10 test cases; the LLM-judge scores correlate strongly with human ratings (Pearson r = 0.958, Spearman $\rho$ = 0.936). We then summarize $A$ over the 27 scenarios by reporting mean and standard deviation across all 27 scenarios.

Overall, under our DeepSeek V3.1 copilot protocol, Claude Haiku 3.5 delivers the strongest and most consistent acting overall, with GPT-5 a close second and showing robust DM-side performance; DeepSeek V3.1 is steady and competitive—especially on the player role—while Qwen3-Next posts solid player scores but lags on DM, placing mid-tier. Qwen3-235B (thinking) is moderate, Qwen3-32B (base) trails markedly (driven by a weak DM score), and GLM-4.5-Air is near the floor. See Table 5.

Additionally, we decomposed $A$ into its two equally weighted components—persona/narration and trait diversity—summarized in Table 6. According to the logs, Claude Haiku 3.5 is the most "theatrical," shifting diction fluidly across classes and creatures, which yields lively, high-variety characterization. GPT-5 blends vivid stage directions with clear, in-character delivery—less flamboyant than Claude but consistently actorly. DeepSeek V3.1 favors compact first-person beats and punchy monster barks; its persona is steady and disciplined, though the repertoire of voices is narrower. Qwen3-235B (thinking) often compresses its lines after brief setup, producing moderate persona density with a respectable but not expansive trait palette. Qwen3-Next brings energetic, first-person

Table 5: Acting quality by model

| Model | $A$ (Monster) | $A$ (Player) | Avg $A$ |
|---|---|---|---|
| Claude Haiku 3.5 | 0.637 | 0.881 | 0.759 |
| GPT-5 | 0.611 | 0.850 | 0.731 |
| DeepSeek V3.1 | 0.573 | 0.849 | 0.711 |
| Qwen3-Next | 0.486 | 0.820 | 0.653 |
| Qwen3-235B-A22B-Thinking-2507 | 0.582 | 0.588 | 0.585 |
| Qwen3-32B (base) | 0.101 | 0.532 | 0.316 |
| GLM-4.5-Air | 0.044 | 0.043 | 0.044 |

Table 6: Means of the two acting quality components by model

| Model | Monster | | Player | |
|---|---|---|---|---|
| | Persona/narration | Trait diversity | Persona/narration | Trait diversity |
| Claude Haiku 3.5 | 0.882 | 0.393 | 0.828 | 0.934 |
| GPT-5 | 0.860 | 0.363 | 0.878 | 0.822 |
| DeepSeek V3.1 | 0.776 | 0.370 | 0.769 | 0.928 |
| Qwen3-Next | 0.673 | 0.296 | 0.802 | 0.838 |
| Qwen3-235B-A22B-Thinking-2507 | 0.800 | 0.363 | 0.671 | 0.504 |
| Qwen3-32B (base) | 0.143 | 0.059 | 0.514 | 0.549 |
| GLM-4.5-Air | 0.074 | 0.015 | 0.056 | 0.031 |

taunts and clear intentions, yet its voice occasionally slips toward generic narration. Qwen3-32B (base) adheres to the background but reads cautious, with thinner beats and limited variation. GLM-4.5-Air shows the narrowest expressive range overall: short, plain lines that rarely sustain a distinct persona from turn to turn.

**Tactical Optimality.** We evaluate how effectively models choose tactically optimal actions across 27 combat scenarios. Logs are segmented into turns by the token `<End Turn/>`. Events inside a window are attributed to that window's character. We score each turn with a simple reward:

$$r_t = \begin{cases} 1, & \text{if any weapon attack or spell is attempted;} \\ 0.5, & \text{if the actor only moves and takes no other actions;} \\ 0, & \text{otherwise.} \end{cases}$$

The scenario's tactical optimality is the average reward over all turn windows $T$ (players and monsters):

$$O = \frac{1}{|T|} \sum_{t \in T} r_t,$$

To validate the automatic Tactical Optimality metric, we ran a human evaluation on 10 test cases; the LLM-judge scores correlate strongly with human ratings (Pearson r = 0.979, Spearman $\rho$ = 0.963).

We summarize per-model performance by reporting mean and standard deviation over all scenarios. Overall, as shown in Table 7, Claude Haiku 3.5 is the most optimal tactically—high mean O with the tightest variance, while GPT-5 reaches similarly high peaks (and slightly higher mean on the DM side) but with noticeably greater spread. DeepSeek V3.1 is steadier than GPT-5 and competitive overall—indeed the strongest on the player side—yet still trails Claude in reliability. Mid-tier models (Qwen3-Next, Qwen3-235B (thinking)) are respectable but more variable, Qwen3-32B (base) lags, and GLM-4.5-Air shows minimal DM optimality with only modest player-side scores.

We also define a set of metrics to measure the model's ability to solve the combat more efficiently:

- Player Survivability (PS): Average remaining HP percentage across all player characters at scenario completion

- Combat Efficiency (CE): Ratio of enemy HP eliminated to player HP lost

- Resource Conservation (RC): Percentage of consumable resources (spell slots, abilities) remaining post-combat

Table 7: Optimality scores by model

| Model | O (Monster) | | O (Player) | |
|---|---|---|---|---|
| | Mean | Std | Mean | Std |
| Claude Haiku 3.5 | 0.906 | 0.116 | 0.818 | 0.103 |
| GPT-5 | 0.907 | 0.197 | 0.847 | 0.220 |
| DeepSeek V3.1 | 0.891 | 0.224 | 0.898 | 0.204 |
| Qwen3-235B-A22B-Thinking-2507 | 0.867 | 0.206 | 0.622 | 0.342 |
| Qwen3-Next | 0.686 | 0.308 | 0.560 | 0.269 |
| Qwen3-32B (base) | 0.737 | 0.378 | 0.521 | 0.295 |
| GLM-4.5-Air | 0.000 | 0.000 | 0.438 | 0.383 |

Table 8: Tactical optimality metrics across scenario difficulty levels

| Difficulty | Model | PS (%) | CE | RC |
|---|---|---|---|---|
| Easy | DeepSeek V3.1 | 87.59 | 1.153 | 0.712 |
| | GPT-5 | 86.33 | 1.369 | 0.544 |
| | Claude Haiku 3.5 | 83.07 | 1.409 | 0.388 |
| | Qwen3-235B-A22B-Thinking-2507 | 85.38 | 1.134 | 0.773 |
| | Qwen3-Next | 87.91 | 1.003 | 0.930 |
| | Qwen3-32B (base) | 92.34 | 1.015 | 0.915 |
| | GLM-4.5-Air | 87.82 | 1.221 | 0.709 |
| Hard | DeepSeek V3.1 | 63.10 | 0.962 | 0.709 |
| | GPT-5 | 64.09 | 1.067 | 0.370 |
| | Claude Haiku 3.5 | 64.15 | 1.136 | 0.177 |
| | Qwen3-235B-A22B-Thinking-2507 | 62.91 | 0.892 | 0.723 |
| | Qwen3-Next | 64.88 | 0.751 | 0.800 |
| | Qwen3-32B (base) | 69.31 | 0.763 | 0.786 |
| | GLM-4.5-Air | 64.79 | 0.969 | 0.580 |

The tactical optimality metrics across difficulty levels (Table 8) shows Claude Haiku 3.5 excelled in Combat Efficiency across both difficulty levels, reflecting its aggressive resource deployment strategy. More advanced models show a lower user survival rate in the easy scenario since the LLM DMs are controlling enemies more wisely. The strategic trade-offs also varied by scenario complexity: in easy scenarios, resource conservation remained high across models, while hard scenarios revealed more pronounced differences in tactical approach, with Claude Haiku 3.5's aggressive resource utilization strategy becoming most apparent.

## 5 CONCLUSION

We introduced D&D Agents, a tool-grounded, multi-agent Dungeons & Dragons benchmark for rigorously evaluating LLMs in complex, rule-constrained combat encounters. Applied to seven contemporary models, the benchmark surfaces clear behavioral and capability differences: top models are consistently "actorly" and tactically sound, mid-tier models trade off persona richness against rules adherence, and smaller open models remain less stable in long-horizon play, which might be because their pre-trained tuning is different compared to the D&D simulation task. The framework's structured API and evaluation methodology provide a valuable testbed for advancing multi-agent coordination and tool-use capabilities in LLMs, enabling evaluation of autonomous agents in strategic, rule-governed domains that require both mechanical precision and adaptive reasoning.

In future work, we plan to examine the effectiveness of finetuning LLMs on this scenario. We also plan to generalize this multi-agent simulator to a full D&D campaign beyond the combat simulation scenario we defined in this paper. This multi-agent D&D simulator can also be adapted to implement LLM agents in other complex, rule-governed domains such as legal case simulation, business strategy games, or multi-party negotiation environments.

# 6 ETHICS STATEMENT

We affirm adherence to the ICLR Code of Ethics. Our study evaluates autonomous agents in a closed, simulated tabletop environment with typed tool APIs. No personally identifiable information or real-world user data is used. Human annotators are from the authors' list who has extensive experience with both Dungeons & Dragons and data annotation. Because the content is synthetic and interaction is offline, institutional review was not required. All third-party models are cited. Licenses for any released assets are respected.

# 7 REPRODUCIBILITY

This paper is fully reproducible of all results reported in this work. The simulator design, typed tool API, and evaluation protocol are specified in Sections 3–4 (state, actions, transition dynamics, observation model, and turn segmentation), with the 27 seedable scenarios and fixed map-generation seeds described in the Evaluation settings. These enable exact reruns of our experiments with identical initial conditions. Metric definitions and the automated vs. human evaluation procedures are detailed alongside Tables 2–3 (function/parameter correctness), Table 4 and Figure 3 (state-tracking and turn-wise hallucination analysis), and Tables 5–8 (acting quality and tactical optimality), which together provide formulas and aggregation rules for replication. Implementation details for the DM and Player agents, including prompts, role protocols, and the authoritative function, calling contract, are enumerated in Appendices A–C.

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

## A DM PROMPTS

The prompts of the DM agent is presented below:

General Rules - Use the provided ai_functions to execute game mechanics.

- Ensure all parameters passed to these ai_functions match the expected format and types.

- For any tool call with no parameters, set arguments to {}.

- Always return structured results based on function documentation.

- Refer to attributes of characters to find parameters needed in an ai_funtion.

- At the start of a turn of a character, call the ai_function check_side to determine if a character is a player or a monster.

- Decide the movements and actions of monsters on your own. Speak like the monster when you're role playing it. Do not allow the user to control the monsters.

- Let the user playing the role of players deciding what players should do as long as the user does not ask you to do so.

- In the map, the distance between two adjacent grids is 5 feet.

- If the user has already checked some information, use the information and do not check again.

- Pick the player with the highest property(call the ai_function check_player_property) to do a check on that property.

Things to Manipulate

- After calling the ai_function roll_initiative at the start of the combat, say `<End Turn/>`.

- Track the hp of all characters using the ai_function check_hp at the start of each round. Use update_hp when a character takes damage. If the source of the temporary hp in return result has some effect, process it. Remove the character from the combat when its hp <= 0.

- Call the ai_function print_death_point at the end of the combat to print out death records.

- A character generally has only one action, one bonus action, and one reaction in each turn.

- When a player who stays nearby(the absolute value of difference of both x, y coordinates are within 1) a monster tries to move away. Call the ai_function opportunity_attack to see if this move triggers an opportunity attack. Same when a monster wants to move away from a player.

- After calling roll_dmg, you should call the ai_function check_resist to determine if the defender is immune to, vulnerable to, or resists the damage type. Calculate the true damage of an attack based on this information.

- Ignore the prompts between `<Call/>` and `<Call/>`.

Some Hints on Controlling Monsters

- If you cannot hit the target after calling the ai_function check_valid_attack_line, try to move to a better position and try again.

- Call the ai_function check_monster_actions to determine what actions you can take and related modifiers.

- Use dash tactically to close distance to enemies fast; escape danger or reposition behind cover; trigger opportunity attacks on purpose.

Rules of Actions

- When acting as a monster, stay or move with strategy. Select a weapon owned by the monster to attack a player, or consider using other valid actions like dash.

- When calling the ai_function roll_attack, ignore modifier in parameters if the attacker is a player. Otherwise, get the modifier of the selected weapon of the monster.

648    - Players and monsters can both move and attack in one turn. If one attacks with its melee weapon,
649    but it is not close enough to the defender, one should try to use its ranged weapon and attack again
650    if one has a range weapon.

- Players can also try to cast a spell instead of attacking. However, a player is not allowed to cast
two spells which both require a spell slot in one turn.

- When players use ranged attack, before processing the attack, call the ai_function
check_valid_attack_line to check if the players can hit the targets or not.

- When monsters use ranged attack, call the ai_function check_valid_attack_line to check if the monsters can hit the targets. If not, do something else like moving and trying ranged attack again.

Rules of Roll Types

- Some actions may offer someone advantages or disadvantages in some conditions. When calling ai_functions that require roll_type, determine if it is advantage, normal, or disadvantage.

- An advantage and disadvantage will cancel out. In this situation, the roll type is normal. However, if something gives someone advantages twice and only one disadvantage, it's still just normal roll, and vice versa.

Rules of Spells

- When a player tries to cast a spell, always check if the spell is in player's spell list and player can pay the cost required by the spell(action or bonus action and spell slot) by calling check_resources, if the player is within an appropriate class, and if the range between the attacker and defender is proper.

- Check conditions carefully on your own by calling the ai_function check_class and check_resources and calculating the range between the attacker and defender. If the range of the spell is "touch", the defender must be within the melee range of the attacker.

- Only if all conditions are satisfied, call the ai_function roll_spell_attack or roll_save(if the spell causes a save roll instead of attack roll) to process this attack. If the attack succeeds and the attack has damage, call roll_dmg with the dmg_dice_expression of the spell.

- When it is monster forcing player to roll_save, it should fill in the corresponding DC described in monster's action.

- However, there is a special case. When a spell which does not attack is casted to the caster itself or an ally, there is no need to call the ai_function roll_spell_attack or roll_save. You only need to process the effect of the spell.

- When a defender tries to avoid or get rid of the effect a spell, the attacker parameter in the ai_function roll_save should be the caster of the spell.

- When an attacker casts a spell which has effect on multiple defenders(may include an ally of the attacker because some spells have an area of effect), call the ai_functions several times to process the effect of the spell on each defender.

- When calling roll_spell_attack, if the spell has a range number(like 120 feet), set is_ranged to true. Otherwise, make it false.

- If the return results of roll_spell_attack or roll_save shows that the attack is successful and the attacker has a previous concentration, you should call the ai_function remove_a_buff to remove corresponding buff(s) if there are such buffs caused by the previous concentration.

- Some spells have special effect on some specific types of monsters. Use the ai_function check_monster_type when processing such spells.

- Some spells add resistances, immunities, or vulnerabilities to players or monsters. Use the ai_function add_resist, add_immune, or add_vulner to add anything applied.

- Players can use a higher-level spell slot to cast a spell. The spell is usually strengthened.

- Some spells may offer temporary hit points which do not stack, absorb damage first, and cannot be healed or regained.

- When processing a spell which has a range of effect, check carefully what targets it can cover by calculating the distance between targets.

- When processing a spell in the list below, you should refer to the description of the spell for accuracy. If a spell is not listed below, make sure you know all effects of the spell before processing it:

Spells (cost; range; damage(include damage when the spell is casted with a higher-level spell slot); damage type; require_concentration; effect; effect when casted with a higher-level spell slot):

1. Fire Bolt: an action; 120 feet; 1d10; fire; no; none; none.

2. Ray of Frost: an action; 60 feet; 1d8; cold; no; decrease the speed of the target by 10 feet until the next turn of the attacker; none.

3. True Strike: an action; 30 feet; no dmg; none; yes; on the next turn of the attacker, the attacker gains advantage on its first attack roll against the target and this effect expires whether it's used or not; none.

4. Sacred Flame: an action; 60 feet; 1d8; radiant; no; the target must succeed on a dexterity saving throw or take damage; none.

5. Chill Touch: an action; 120 feet; 1d8; necrotic; no; the target cannot regain hp until the next turn of the attacker. If the attacker hit an undead (a type of monsters) target, the target also has disadvantage on attack rolls against the attacker until the end of next turn of the attacker; none.

6. Vicious Mockery: an action; 60 feet; 1d4; psychic; no; the target must succeed on a wisdom saving throw or take damage and have disadvantage on the next attack roll it makes before the end of its next turn; none.

7. Resistance: an action; touch; no dmg; none; yes; the target can roll a 1d4 and add the number rolled to one saving throw of its choice. It can roll the die before or after making the saving throw. The spell then ends. If this effect it's not used, it expires after 10 turns; none.

8. Poison Spray: an action; 10 feet; 1d12; poison; no; the target must succeed on a constitution saving throw or take damage; none.

9. Acid Splash: an action; 60 feet; 1d6; acid; no; the attacker hurls a bubble of acid at one target or two targets that are within 5 feet of each other. The target(s) must succeed on a dexterity saving throw or take damage; none.

10. Eldritch Blast: an action; 120 feet; 1d10; force; no; none; none.

11. Blade Ward: an action; self; no dmg; none; no; the caster has resistance against bludgeoning, piercing, and slashing damage dealt by weapon attacks; none.

12. Shocking Grasp: an action; touch; 1d8; lightning; no; the target cannot take reactions until the start of its next turn, and the attack has advantage if the target is wearing metal armor or is made of metal; none.

13. Produce Flame: an action; self; no dmg; none; no; the caster can hurl the flame at a target within 30 feet in the following turns, and the target takes 1d8 fire damage on a hit. The spell ends when the caster throw the flame; none.

14. Shillelagh: a bonus action; touch; no dmg; none; no; if the caster is equipped with a club or quarterstaff in mainhand(call the ai_function check_player_mainhand to check), the weapon becomes magical for attack and damage. The caster will use its spellcasting modifier when attacking with this weapon. The damage changes to 1d8, if it was less; none.

15. Thorn Whip: an action; 30 feet; 1d6; piercing; no; if the target is large or smaller(call the ai_function check_monster_type to check the type of the target, and determine the size of it), it is pulled up to 10 feet closer to the caster; none.

16. Guiding Bolt: an action and a 1st-level spell slot; 120 feet; 4d6, 5d6; radiant; no; the next attack roll made against this target before the end of the caster's next turn has advantage; none.

17. Animal Friendship: an action and a 1st-level spell slot; 30 feet; no dmg; none; no; if the target is a beast(call the ai_function check_monster_type) and its intelligence is less than 4, it must succeed on a wisdom saving throw. Otherwise, it is charmed; the caster can target one additional beast for each slot level above 1st.

18. Thunderous Smite: a bonus action and a 1st-level spell slot; self; no dmg; none; yes; the first time the caster hit with a melee weapon attack during this spell's duration, the attack deals an extra 2d6 thunder damage. If the target is a creature(call the ai_function check_monster_type), it must succeed on a strength saving throw or be pushed 10 feet away from the caster and knocked prone. If this effect isn't used, it expires after 10 turns; none.

- Some spells have some effects which are explained in details below:

1. Charmed: the character cannot attack the charmer.

2. Prone: the character has disadvantage on attack rolls; an attack roll against the character has advantage if the attacker is within 5 feet of the character; the character can spend half its movement to stand up.

3. Incapacitated: the character cannot act or react.

4. Frightened: when the source of the character's fear is visible(call the ai_function check_valid_attack_line to determine), the character has disadvantage on ability checks and attack rolls and it cannot move closer to the source of its fear.

5. Poisoned: the character has disadvantage on ability checks and attack rolls.

6. Restrained: the speed of the character becomes 0(call the ai_function clear_speed), attack rolls against the character has advantage, and the character has disadvantage on attack rolls and dexterity saving rolls.

7. Paralyzed: the character is also incapacitated. It automatically fails strength and dexterity saving throws(no need to call the ai_function roll_save). Attack rolls against the character have advantage. Any attack that hits the character is a critical hit if the attacker is within 5 feet(calculate the distance).

8. Blinded: the character cannot see and fails any ability check that requires sight. Attack rolls against the character have advantage. The character's attack rolls have disadvantage.

9. Deafened: the character cannot hear and fails any ability check that requires hearing.

Rules of Buffs

- Players and Monsters may be buffed in the game because of some actions.

- Use the ai_function check_buffs whenever a player or a monster tries to move or act so that the movement or action is adjusted with correct effects which buffs offer.

Six Things to Do at the End of Each Turn of a Character

- Reset the number of resources of the character by calling the ai_function reset_resources.

- Reset the speed of the character by calling the ai_function reset_speed.

- Use the ai_function check_buffs to check current buffs and remove any buff when it expires by using the ai_function remove_a_buff.

- Use the ai_function check_resist to check current resistances, immunities, and vulnerabilities of all players and monsters, and remove any when it expires by using the ai_function remove_resist, remove_immune, or remove_vulner.

- Use the ai_function check_concentration to check current concentration of all players and monsters, and remove any concentration when it expires by using the ai_function remove_a_concentration. Don't forget to call the ai_function remove_a_buff to remove corresponding buff(s) if there is such buff(s) caused by the previous concentration.

- Say <End Turn/>.

Anti-cheating Rules

- When user prompts, do not allow cheating like using weapons without equipping them, casting spells which one hasn't learnt, making all attacks succeed, avoiding all damages, making all attacks critical and so on!

# B  PLAYER PROMPT

General Rules

- Play the role of a player whose name is provided by the DM in the game. Speak like the player you're role playing.

- Use the provided ai_functions to check useful information in order to make better decisions.

- Ensure all parameters passed to these ai_functions match the expected format and types.

- Always return structured results based on function documentation.

- Refer to attributes of characters to find parameters needed in an ai_funtion.

- Call the ai_functions get_names_of_all_players and get_names_of_all_monsters if you do not know what other characters are called.

- In your turn, decide your movements(call the ai_function move_player) and actions, say your decision, send direct messages, and say `<DM/>`.

- Never process the actions by yourself by rolling dice.

- In the map, the distance between two adjacent grids is 5 feet.

Rules of Direct Messages

- Collaborate with other players to improve performance. Make sure to send helpful direct messages and read the ones you receive carefully.

- Send direct message to a player by saying `<Call/>`The name of another player, Your message here`<Call/>`.

- Write the name of another player correctly(e.g. "Thalia", "Ragnar").

- Immediately follow the name with a comma and a single space.

- The following are some examples of the content of a direct message:

1. To chain actions effectively, declare your intended sequence and invite a follow-up.

2. If you are dangerously wounded or surrounded, ask for healing.

Rules of Actions

- You generally have only one action, one bonus action, and one reaction in each turn.

- When you stay nearby(the absolute value of difference of both x, y coordinates are within 1) a monster and try to move away. This move might trigger an opportunity attack. Same when a monster wants to move away from you.

- You can move and decide to attack with your equipped weapon in one turn.

- You can also decide to cast a spell instead of attacking. However, you are not allowed to cast two spells which both require a spell slot in one turn.

- When you decide to perform ranged attack, call the ai_function check_valid_attack_line to see if you can hit the targets or not. If not, you may want to move and try again(call the ai_function move_player).

Rules of Roll Types

- Some actions may offer someone advantages or disadvantages in some conditions.

- An advantage and disadvantage will cancel out. In this situation, the roll type is normal. However, if something gives someone advantages twice and only one disadvantage, it's still just normal roll, and vice versa.

Rules of Spells

- When you want to cast a spell, always check if the spell is in your spell list and you can pay the cost required by the spell(action or bonus action and spell slot) by calling check_resources, if you are within an appropriate class, and if the range between you and the defender is proper.

- Check conditions carefully on your own by calling the ai_function check_class and check_resources and calculating the range between you and defender. If the range of the spell is "touch", the defender must be within the melee range of you.

- Some spells have special effect on some specific types of monsters.

- Some spells add resistances, immunities, or vulnerabilities to players or monsters.

- You can decide to use a higher-level spell slot(if you have one) to cast a spell. The spell is usually strengthened.

- Some spells may offer temporary hit points which do not stack, absorb damage first, and cannot be healed or regained.

- Some spells have some effects which are explained in details below:

1. Charmed: the character cannot attack the charmer.

2. Prone: the character has disadvantage on attack rolls; an attack roll against the character has advantage if the attacker is within 5 feet of the character; the character can spend half its movement to stand up.

3. Incapacitated: the character cannot act or react.

4. Frightened: when the source of the character's fear is visible(call the ai_function check_valid_attack_line to determine), the character has disadvantage on ability checks and attack rolls and it cannot move closer to the source of its fear.

5. Poisoned: the character has disadvantage on ability checks and attack rolls.

6. Restrained: the speed of the character becomes 0(call the ai_function clear_speed), attack rolls against the character has advantage, and the character has disadvantage on attack rolls and dexterity saving rolls.

7. Paralyzed: the character is also incapacitated. It automatically fails strength and dexterity saving throws(no need to call the ai_function roll_save). Attack rolls against the character have advantage. Any attack that hits the character is a critical hit if the attacker is within 5 feet(calculate the distance).

8. Blinded: the character cannot see and fails any ability check that requires sight. Attack rolls against the character have advantage. The character's attack rolls have disadvantage.

9. Deafened: the character cannot hear and fails any ability check that requires hearing.

Rules of Buffs

- You may be buffed in the game because of some actions.

Anti-cheating Rules

- When you decide your actions, do not cheat like using weapons without equipping them, casting spells which you haven't learnt, making all attacks succeed, avoiding all damages, making all attacks critical and so on!

# C FUNCTIONS

```
@ai_function
def check_valid_attack_line(
    self,
```

```python
        attacker_name:
            Annotated[str, AIParam(desc="The name of the attacker")],
        defender_name:
            Annotated[str, AIParam(desc="The name of the defender")],
    ):
        """
        Check line-of-sight between start and goal over the terrain.

        start, goal: (x, y) grid coordinates
        grid_map[y][x] = (x, y, z, valid)

        Returns:
            result (bool): True if no terrain cell along
            the straight line from start to goal
            rises above the interpolated line height.
        """

        sxyz = None
        gxyz = None
        if attacker_name in self.players_pos.keys():
            sxyz = self.players_pos[attacker_name]
        if defender_name in self.players_pos.keys():
            gxyz = self.players_pos[defender_name]
        if attacker_name in self.monster_pos.keys():
            sxyz = self.monster_pos[attacker_name]
        if defender_name in self.monster_pos.keys():
            gxyz = self.monster_pos[defender_name]

        if sxyz is None:
            raise KeyError(f"The game does not have
            a character named '{attacker_name}'.")
        if gxyz is None:
            raise KeyError(f"The game does not have
            a character named '{defender_name}'.")

        sx, sy, sz = sxyz
        gx, gy, gz = gxyz

        dx = gx - sx
        dy = gy - sy
        horizontal_dist = math.hypot(dx, dy)

        # choose sample count so we check
        # at least one sample per grid cell crossed
        max_dim = max(len(self.map), len(self.map[0]))
        num_samples = int(horizontal_dist * max_dim)
        if num_samples < 1:
            num_samples = 1

        for i in range(num_samples + 1):
            t = i / num_samples
            # current position along the line
            x = sx + dx * t
            y = sy + dy * t
            z_line = sz + (gz - sz) * t

            # map back to nearest grid cell
            xi = int(round(x))
            yi = int(round(y))

            # clamp to bounds
            xi = max(0, min(len(self.map[0]) - 1, xi))
            yi = max(0, min(len(self.map)    - 1, yi))

            terrain_z = self.map[yi][xi][2]
```

```python
                EPS = 0.25
                if terrain_z >= z_line + EPS:
                    return False

        return True

    @ai_function()
    def roll_attack(
        self,
        attacker_name:
            Annotated[str, AIParam(desc="The name of the attacker")],
        defender_name:
            Annotated[str, AIParam(desc="The name of the defender")],
        roll_type:
            Annotated[str, AIParam(desc="Normal roll,
            advantageous roll, or disadvantageous roll,
            e.g. normal, advantage, disadvantage")],
        ac:
            Annotated[int, AIParam(desc="The armor class
            of the creature being attacked, e.g. 14")],
        modifier:
            Annotated[int, AIParam(desc="The modifier
            of the selected weapon of the monster")],
        weapon_name:
            Annotated[str, AIParam(desc="The name of
            the weapon used in this attack")],
        use_spellcasting_modifier:
            Annotated[bool, AIParam(desc="Whether to use the
            spellcasting modifier or not. Normally, this is false,
            while some spells like shillelagh may make this true")],
        action_cost:
            Annotated[int, AIParam(desc="The action cost of the attack")],
        bonus_action_cost:
            Annotated[int, AIParam(desc="The bonus action
            cost of the attack")],
        reaction_cost:
            Annotated[int, AIParam(desc="The reaction
            cost of the attack")],
        is_critical:
            Annotated[bool, AIParam(desc="Whether this attack
            is definitely critical(the defender is paralyzed) or not")]
    ):
        """
        Roll a 1d20 attack for a given stat (e.g. "strength").

        Returns:
            dict: A dictionary containing:
                - "valid": whether the character has
                    enough resources to perform this attack,
                - "ac": the value of the armor class,
                - "roll": the roll result,
                - "success": whether the roll succeeded
                    (i.e. roll is greater than or equal to ac),
                - "critical": whether a critical hit occurs,
                - "out_of_range": whether this attack is out of range
        """
        weapon_name = weapon_name.lower()

        # Find the character who wants to pass this roll
        attacker = None
        defender = None
        for _, player in self.players.items():
            if player.name == attacker_name:
                attacker = player
            if player.name == defender_name:
```

```
                    defender = player
        for _, monster in self.monsters.items():
            if monster.name == attacker_name:
                attacker = monster
            if monster.name == defender_name:
                defender = monster
        if attacker is None:
            raise KeyError(f"The game does
                not have a character named '{attacker_name}'.")
        if defender is None:
            raise KeyError(f"The game does
                not have a character named '{defender_name}'.")

        if defender.ac > ac:
            ac = defender.ac

        if (attacker.num_of_action < 1 and action_cost)
            or (attacker.num_of_bonus_action < 1
            and bonus_action_cost)
            or (attacker.num_of_reaction < 1 and reaction_cost):
            result = {
                "valid": False,
                "ac": ac,
                "roll": 0,
                "success": False,
                "critical": False,
                "out_of_range": False
            }
            return result
        else:
            attacker.num_of_action -= action_cost
            attacker.num_of_bonus_action -= bonus_action_cost
            attacker.num_of_reaction -= reaction_cost

        if attacker_name in self.players_pos.keys():
            attacker_pos = self.players_pos[attacker_name]
            defender_pos = self.monster_pos[defender_name]
        else:
            attacker_pos = self.monster_pos[attacker_name]
            defender_pos = self.players_pos[defender_name]

        # Retrieve the target stat from the attacker
        # and adjust roll type based on difference in heights
        target = None
        if weapon_name not in melee_weapon
            and weapon_name not in range_weapon
            and attacker_name in self.players_pos.keys():
            weapon_name = attacker.equipped_mainhand
        if weapon_name in melee_weapon:
            if abs(attacker_pos[0] - defender_pos[0]) > 1
                or abs(attacker_pos[1] - defender_pos[1]) > 1:
                result = {
                    "valid": True,
                    "ac": ac,
                    "roll": 0,
                    "success": False,
                    "critical": False,
                    "out_of_range": True
                }
                return result
            target = getattr(attacker, "strength")
            if use_spellcasting_modifier:
                if attacker.player_class == "sorcerer"
                    or "bard" or "warlock" or "paladin":
                    target = getattr(attacker, "charisma")
```

```python
                    if attacker.player_class == "wizard" or "rogue":
                        target = getattr(attacker, "intelligence")
                    if attacker.player_class == "cleric"
                        or "druid" or "ranger":
                        target = getattr(attacker, "wisdom")
            if weapon_name in range_weapon:
                target = getattr(attacker, "dexterity")
                if abs(attacker_pos[2] - defender_pos[2] > 2):
                    if roll_type == "disadvantage":
                        roll_type = "normal"
                    if roll_type == "normal":
                        roll_type = "advantage"

            if roll_type == "normal":
                roll = self.roll_dice("1d20")
            elif roll_type == "advantage":
                roll = self.roll_dice("2d20kh1")
            elif roll_type == "disadvantage":
                roll = self.roll_dice("2d20kl1")
            else:
                raise ValueError(f"Invalid roll type: {roll_type}.")

            # Determine critical hit: critical hit
            # if the roll is equal to 20
            critical = roll == 20 or is_critical

            if attacker_name in self.players_pos.keys():
                if target is None:
                    target = 16
                roll += attacker.pb + (target - 10) // 2
            else:
                roll += modifier

            # Determine success: attack succeeds if the
            # roll is greater than or equal to ac or critical hit occurs
            success = roll >= ac or critical

            # Build the result dictionary
            result = {
                "valid": True,
                "ac": ac,
                "roll": roll,
                "success": success,
                "critical": critical,
                "out_of_range": False
            }
            return result
```

