# OpenReview forum: "Setting the DC: Tool-Grounded D\&D Simulations to Test LLM Agents"
_ICLR.cc/2026/Conference — ICLR 2026 Conference Withdrawn Submission_

### Official Review · Reviewer_jwEZ · 2025-10-26

**Soundness:** 3
**Presentation:** 3
**Contribution:** 3
**Rating:** 2
**Confidence:** 4

**Summary:**

This paper introduces D&D Agents, a benchmark for evaluating large language models in multi-agent Dungeons & Dragons combat simulations. The authors develop a simulator that enables LLMs to play as Dungeon Master, players, and monsters through structured API calls. The system evaluates model performance across six dimensions: Function Usage, Parameter Fidelity, Acting Quality, Tactical Optimality, State Tracking, and Function Efficiency. Testing seven contemporary models on 27 reproducible combat scenarios, the authors find that models like Claude Haiku 3.5 and GPT-5 demonstrate superior performance in rule adherence and tactical decision-making, while smaller models struggle with long-horizon planning and state tracking.

**Strengths:**

The paper makes a genuine attempt to tackle the challenging problem of evaluating LLM agents in long-horizon, rule-governed scenarios. The engineering effort to create a complete D&D combat simulator with proper rule enforcement deserves recognition, as does the attention to reproducibility through fixed seeds and deterministic mechanics. The six-dimensional evaluation framework thoughtfully captures different aspects of agent behavior beyond simple success metrics, and the validation of automated metrics against human judgment demonstrates methodological rigor in that specific component.The benchmark successfully enables fair comparison across multiple models under identical conditions, which has clear practical value for researchers studying tool use and planning. The inclusion of both closed and open-source models in the evaluation provides useful data points about current capabilities. The detailed prompts and function specifications in the appendices genuinely support reproducibility, and researchers could conceivably use this framework to test new models or prompting strategies.

**Weaknesses:**

The paper fails to articulate why D&D specifically is valuable as an AI benchmark compared to established alternatives like StarCraft II, Minecraft, or Diplomacy. Each game environment tests distinct capabilities—StarCraft requires real-time strategic planning under partial observability, Minecraft tests open-ended exploration and tool use, Diplomacy emphasizes negotiation and theory of mind. The paper never explains what unique AI challenges D&D poses beyond these existing benchmarks. Moreover, by restricting evaluation to combat only, the paper eliminates precisely the capabilities that would distinguish D&D: creative problem-solving in exploration, improvisation in social encounters, and narrative coherence across diverse scenarios. What remains is essentially a turn-based tactical combat simulator that could be instantiated in many settings beyond D&D.
The comparison with existing D&D work is inadequate. FIREBALL contains nearly 25,000 real gameplay sessions with complete state tracking from actual human players. CALYPSO was deployed with 71 real players engaging in exploration, social interaction, and combat. The paper claims these are not "closed-loop multi-agent simulations" but provides no substantive technical distinction. Both systems involve LLMs generating structured actions that a game engine executes with state feedback. The authors assert their system is novel because LLMs "execute" mechanics while prior work only "advises," but this distinction collapses when recognizing both approaches use hardcoded game engines processing LLM-generated inputs. The paper needs explicit comparison demonstrating what technical capability exists here that Avrae-based systems fundamentally cannot provide.
The copilot protocol fatally compromises individual model evaluation. When Model X plays as DM against DeepSeek-controlled players, X's measured performance conflates its own capabilities with DeepSeek's tactical choices. The paper provides no analysis of how copilot identity affects results, no comparison of Model X with different copilot partners, and no justification for why this confounded evaluation is preferable to alternatives like self-play or human-AI collaboration. Table 1 shows interesting failure modes where models check conditions then violate them, but the paper never investigates why. Without mechanistic analysis of what causes these failures—context length limitations, training data gaps, or fundamental issues in tool-use learning—the benchmark becomes merely a leaderboard rather than a research tool advancing our understanding.

**Questions:**

Why is D&D superior to existing game-based benchmarks for evaluating LLM agents? What specific AI capabilities does D&D test that StarCraft II, Minecraft, or Diplomacy do not adequately assess? Given that this implementation evaluates only combat, how does the remaining challenge differ from generic turn-based strategy games?
What technical capabilities does this system provide that existing D&D frameworks like Avrae, FIREBALL, or CALYPSO fundamentally lack? Can you provide a comparison table showing specific features present in your system but absent or impossible in prior work? The claim that previous systems are not "closed-loop multi-agent simulations" requires substantiation with concrete technical distinctions.
How do results change with different copilot models? If GPT-5 is evaluated with Claude as copilot instead of DeepSeek, do relative rankings remain stable? What analysis justifies the copilot protocol over self-play or human collaboration alternatives?
Why were the other two pillars of D&D—exploration and social interaction—excluded? Including only combat is like creating a Minecraft benchmark testing only crafting recipes. What prevents extending the framework to scenarios requiring negotiation, puzzle-solving, or improvisation?
Can you provide ablation studies on prompt components and diagnostic analysis of failure modes? When models check line-of-sight, receive False, then attack anyway, what causes this behavior? These mechanistic insights would distinguish scientific contribution from engineering demonstration.

---

### Official Review · Reviewer_zynw · 2025-10-31

**Soundness:** 3
**Presentation:** 3
**Contribution:** 3
**Rating:** 4
**Confidence:** 4

**Summary:**

This paper introduces a benchmark that evaluates how good are LLMs at playing the game Dungeons and Dragons. It first sets up a simulation environment, enabling LLMs to play the game and then use that environment for evaluation. It benchmarks LLMs from different perspectives, including Function Usage, Parameter Fidelity, Acting Quality, Tactical Optimality, State Tracking, and Function Efficiency, trying to evaluate LLMs on strategic gameplay, instruction following and hallucination, etc.

**Strengths:**

1. The paper provides an environment which allows identical seeded game run of D&D. This can serve as a good testbed for other following works
2. It also tries addressing different perspectives of evaluation, not only in strategic gameplay, but also instruction following and hallucination
3. Experiments are conducted on a few SOTA LLMs, and the results show differentiation among them, and indicate a room for further improvement

**Weaknesses:**

1. Lack of examples. Introducing an example that illustrates how a game develops as different players take actions may help.
2. How this work differs from other benchmarks of strategic gameplay (e.g., Avalon and Werewolf) remains unclear.
3. The design of Tactical Optimality seems a bit naive. For example, there might be different spells—some of them are optimal while others are not. However, in the current design of Tactical Optimality, it does not appear to differentiate between different scenarios.

**Questions:**

1. Based on my understanding, D&D is not only a game about strategic gameplay; the quality of the storyline created by the players is also an important aspect. Is it possible to include that as a perspective for benchmarking? For example, could we benchmark LLMs playing as Dungeon Masters (DMs)?

---

> ### Author Response · Authors · 2025-12-04
>
> 1. We appreciate this suggestion. In the revision, we will add a concrete, step-by-step example illustrating how a combat scenario unfolds over several turns as different players and monsters act. The new example will show (i) the natural-language decisions of the DM and players, (ii) the corresponding tool calls (e.g., movement, line-of-sight checks, attacks/spells), and (iii) how these calls update the underlying state (HP, positions, status effects) over time. This will complement the existing high-level description of the simulator and metrics by giving readers a more intuitive, “from-the-log” view of how a full game segment develops in practice.
>
> 2. We agree that a more explicit comparison to other strategic game benchmarks would improve clarity. While Avalon and Werewolf focus on social deduction with hidden roles and primarily linguistic actions (arguments, accusations, voting), our benchmark targets tool-grounded, spatially explicit tactical combat where agents must respect a typed action API, resource budgets, and line-of-sight/range constraints. In the revised version, we will add a comparison table and accompanying text that contrast D&D Agents with Avalon/Werewolf and other multi-agent benchmarks along key axes: action space (typed tools vs. discrete votes), observability (partial but tool-queryable state vs. hidden roles), mechanics (combat and resource management vs. persuasion), and evaluation signals (tool traces and combat outcomes vs. win/loss of a social deduction round). This will highlight the unique role of D&D Agents as a testbed for long-horizon, rules-constrained tool use, complementary to existing social-deduction benchmarks.
>
> 3. We acknowledge that the current Tactical Optimality definition—assigning the same reward to any attack or spell and a smaller reward to “move-only” turns—is intentionally simple and does not differentiate between high-impact and low-impact actions (e.g., strong AoE spells vs. inefficient single-target options). Our goal was to capture a coarse notion of “willingness to engage” that can be computed automatically and that we validate against human ratings (Pearson r ≈ 0.98), but we agree this can be refined. In the revision, we will (i) make this design choice and limitation explicit, (ii) more clearly connect Tactical Optimality to the additional outcome-based metrics we already compute—Player Survivability, Combat Efficiency, and Resource Conservation—which are sensitive to how effective spells and attacks actually are, and (iii) extend the Tactical Optimality metric to incorporate action quality, for example by weighting turns based on realized damage dealt, lethal or disabling effects, and beneficial buffs/heals, rather than treating all attacks/spells as equivalent. This richer scoring function will better distinguish between tactically strong and weak spell/weapon choices while preserving the automatic, log-based nature of our evaluation.
>
> For the question, we agree that narrative quality and Dungeon Master (DM) behavior are central to the full D&D experience, and incorporating them would provide an important additional perspective on LLM capabilities. In this work, however, we scoped the benchmark to tactical combat and treated the DM primarily as a rule-enforcing environment agent. This focus was driven by resource constraints. We see extending our framework to benchmark extensions like storytelling as a natural next step.

---

### Official Review · Reviewer_YPMv · 2025-11-01

**Soundness:** 2
**Presentation:** 1
**Contribution:** 1
**Rating:** 2
**Confidence:** 4

**Summary:**

This paper presents D&D Agents, a multi-agent D&D 5e combat simulator for testing LLMs as Dungeon Master, players, and monsters. Using a typed tool API for state queries, movement, attacks, and dice rolls, agents plan and act in seeded, auditable scenarios. It evaluates seven LLMs across 27 fixed combat encounters on six metrics—function use, parameter accuracy, acting quality, tactical optimality, state tracking, and efficiency—revealing strengths in top models (Claude Haiku 3.5, GPT-5) and role-specific weaknesses.

**Strengths:**

The use of Dungeons & Dragons—a complex, turn-based, and rule-intensive game—as a testbed for LLM agents is interesting.

**Weaknesses:**

1. The benchmark focuses solely on combat encounters from a single D&D module (“Lost Mine of Phandelver”), overlooking other core aspects such as exploration, role-playing dialogue, and puzzle-solving that are integral to full D&D campaigns.
2. The paper does not clearly articulate how this benchmark differs from other text-based simulation environments, such as the Minecraft simulator [1] or the StarCraft II simulator [2].
3. Although the results tables are extensive, the discussion lacks depth. For instance, Tables 7–8 show Claude Haiku outperforming GPT-5, but the paper does not analyze whether this arises from differences in language style, reasoning approach, or model architecture.
4. While the paper mentions the possibility of human–AI co-play, no human baseline experiments are actually included.
5. The writing is somewhat verbose and reads more like an engineering report than an academic paper.



[1] Voyager: An Open-Ended Embodied Agent with Large Language Models. TMLR

[2] Large language models play starcraft ii: Benchmarks and a chain of summarization approach. NeurIPS 2024

**Questions:**

Please refer to Weaknesses.

---

> ### Author Response · Authors · 2025-12-04
>
> 1. We agree that full D&D campaigns include exploration, social role-playing, and puzzle-solving, and that our current benchmark does not aim to cover the entire space of D&D play. Our design choice to focus on tactical combat was intentional because combat provides a rules-dense setting with a clearly defined action economy, spatial constraints, and stochastic outcomes that are well-suited for auditable evaluation of long-horizon behavior.
>
> 2. We agree that the current related-work discussion does not clearly isolate what D&D Agents uniquely contribute relative to other simulation benchmarks like Voyager (Minecraft) and TextStarCraft II. In the revision, we will add a focused comparison section (including a table) that contrasts these environments along key evaluation axes. We will emphasize that D&D Agents differ by placing LLMs in explicit asymmetrical multi-agent roles rather than primarily single-agent control. We are also enforcing a typed, precondition-checked action API where mechanics are executed through tools and narration is non-authoritative, producing auditable traces. It also supports seeded, turn-based replay with explicit action/bonus/reaction budgets and resource accounting.
>
> 3. We agree that the current discussion does not fully unpack the patterns shown in Tables 7–8, especially the cases where Claude Haiku outperforms GPT-5. In the revision, we will expand the analysis section to more carefully examine these differences. Concretely, we will (i) analyze representative transcripts and tool-use traces to compare language style (e.g., verbosity, explicit planning), (ii) characterize differences in reasoning approach (e.g., depth of chain-of-thought, adherence to rules, and error modes), and (iii) discuss how known architectural and training differences between models might plausibly relate to their behavior on our benchmark.
>
> 4. We considered including informal “author-as-player” runs as a human baseline, since several of the authors are experienced D&D players. However, we decided against reporting these as quantitative baselines because they would be (i) extremely low-N, (ii) biased (the authors know the scenarios and evaluation criteria), and (iii) not representative of typical human players. We believe that a meaningful human baseline requires a more systematic study with multiple independent participants, controlled instructions, and standardized skill levels. We will clarify this rationale in the paper and expand the discussion of how our current framework can be directly used to run such controlled human–AI co-play studies in future work.
>
> 5. We appreciate this comment and will revise the paper to make the presentation more concise and aligned with academic writing conventions. Concretely, we will (i) streamline sections that currently contain implementation-level detail, moving some of the lower-level engineering descriptions (e.g., configuration details, deployment, or logging specifics) to an appendix, and (ii) tighten the prose in the main text to focus more on the core research questions, methodology, and findings. We will also revise the introduction and related-work sections to improve the narrative flow and explicitly highlight the conceptual contributions of our benchmark, so that the paper reads less like a system report and more like an academic study of LLM agents in a controlled tactical environment.

---

### Official Review · Reviewer_mQiP · 2025-11-01

**Soundness:** 2
**Presentation:** 1
**Contribution:** 1
**Rating:** 2
**Confidence:** 5

**Summary:**

This paper introduces a closed-loop D&D combat benchmark where LLMs control all roles via structured function calls, enforcing rule compliance and long-horizon coordination. It assesses tool usage, rule fidelity, role-playing consistency, and tactical soundness. Results show larger models excel in execution and narration, while smaller ones struggle with context-heavy DM duties, highlighting challenges in memory, planning, and multi-agent reliability.

**Strengths:**

The paper presents a well-structured multi-dimensional evaluation framework, covering aspects such as Function Usage, Parameter Fidelity, Acting Quality, Tactical Optimality, State Tracking, and Function Efficiency.

**Weaknesses:**

1. With only 27 scenarios (a 3×3×3 design of party compositions, stat tiers, and monster sets), the benchmark may risk overfitting to specific configurations and might not fully capture the inherent randomness of the D&D environment.

2. The paper does not clearly differentiate its benchmark from existing baselines such as ALFWorld, WebArena, or ScienceWorld, which also evaluate LLM agents in interactive or embodied settings.

3. While dice rolls introduce stochasticity, the paper does not analyze performance variance across multiple runs per scenario or discuss how rare events (e.g., critical failures) are handled.

4. The writing and presentation lack rigor and professionalism; for instance, the inclusion of eight tables is odd, and they lack clear descriptions.
 [1] ALFWorld: Aligning Text and Embodied Environments for Interactive Learning.
 [2] WebArena: A Realistic Web Environment for Building Autonomous Agents.
 [3] ScienceWorld: Is Your Agent Smarter than a 5th Grader？

**Questions:**

Questions: Please refer to Weaknesses.

---

> ### Author Response · Authors · 2025-12-04
>
> 1. We agree there aren't enough test cases for the experiments. In order to improve this, we can try to add scriptable success conditions for each D&D scenario (win condition, survival thresholds, objective completion), and report human vs model. The test case can then be reformed by a mix of generator results, i.e., a randomly generated combat with a CR-based generator, including a synthetic monster combination and a player combination with the same parameter control (high, mid, low) with a set of human-curated test sets, i.e., the 3x3x3 now.
>
> 2. We agree that our current framing understates how D&D Agents complement these environments. In the revision, we will add a structured comparison highlighting that D&D Agents differ by explicitly benchmarking multi-agent role structure rather than primarily single-agent task completion, enforcing a typed, precondition-checked action API that cleanly separates narration from mechanics and produces auditable tool traces for reliability analysis, and emphasizing turn-based action economy and resource accounting (action/bonus/reaction, spell slots, conditions) as first-class constraints.
>
> 3. We agree that single-run evaluation is insufficient in a stochastic environment. We will evaluate each scenario under multiple dice seeds and report variance-aware statistics for each metric, as well as rank stability across reruns. We will rerun each fixed initial state for K dice seeds to separate policy quality from luck, include robustness metrics that summarize tail behavior, and explicitly analyze rare-event sensitivity (e.g., critical hits/misses, repeated failed saves) by conditioning performance on these events.

---

### Note · Authors · 2026-01-23

I have read and agree with the venue's withdrawal policy on behalf of myself and my co-authors.